# Breaking Dynamic Behavior in 3D Covalent Organic Framework with Pre-Locked Linker Strategy

**DOI:** 10.3390/nano14040329

**Published:** 2024-02-07

**Authors:** Xiaohong Chen, Chengyang Yu, Yusran Yusran, Shilun Qiu, Qianrong Fang

**Affiliations:** 1College of Chemistry, State Key Laboratory of Inorganic Synthesis and Preparative Chemistry, Jilin University, Changchun 130012, China; 2College of Chemistry and Environmental Engineering, Wuhan Polytechnic University, Wuhan 430023, China

**Keywords:** covalent organic framework, breaking dynamic behavior, pre-locked linker, gas storage

## Abstract

Due to their large surface area and pore volume, three-dimensional covalent organic frameworks (3D COFs) have emerged as competitive porous materials. However, structural dynamic behavior, often observed in imine-linked 3D COFs, could potentially unlock their potential application in gas storage. Herein, we showed how a pre-locked linker strategy introduces breaking dynamic behavior in 3D COFs. A predesigned planar linker-based 3,8-diamino-6-phenylphenanthridine (DPP) was prepared to produce non-dynamic 3D JUC-595, as the benzylideneamine moiety in DPP locked the linker flexibility and restricted the molecular bond rotation of the imine linkages. Upon solvent inclusion and release, the PXRD profile of JUC-595 remained intake, while JUC-594 with a flexible benzidine linker experienced crystal transformation due to framework contraction–expansion. As a result, the activated JUC-595 achieved higher surface areas (754 m^2^ g^−1^) than that of JUC-594 (548 m^2^ g^−1^). Furthermore, improved CO_2_ and CH_4_ storages were also seen in JUC-595 compared with JUC-594. Impressively, JUC-595 recorded a high normalized H_2_ storage capacity that surpassed other reported high-surface area 3D COFs. This works shows important insights on manipulating the structural properties of 3D COF to tune gas storage performance.

## 1. Introduction

Covalent organic frameworks (COFs) are an emerging class of crystalline porous polymers prepared by linking predesigned organic building units through covalent bonds [1,2,3,4,5,6]. A wide range of organic building units with a specific chemical composition and functionality can be incorporated into COF structures to afford functional materials [7,8,9]. As this material is composed of periodical skeletons and an ordered nanopore structure, COFs exhibit low density and high porosity, making them attractive for gas storage and separation applications. Furthermore, based on the dimension of the building units, both two-dimensional (2D) COFs with a laminar structure and 3D COFs with extended 3D networks can be designed and synthesized [10,11]. Compared to 2D COFs, 3D COFs are far more excellent for gas storage due to their higher surface areas and capability to load more active sorption sites [12,13,14,15,16]. However, due to the inexistence of interlayer π-π bonding in 3D structures, 3D COFs often experience structural dynamic resulting in structural dynamic expansion and contraction upon guest absorption, release, and exchanges [17]. While this structural characteristic can be beneficial for size-dependent molecular separation, it crucially impedes the gas storage capacity of 3D COFs as it causes a reduction in surface areas and pore size, as well as the pore volume of the 3D COFs. For instance, LZU-301, governed by flexible imine linkages and a bipyridine linker, experienced framework contraction after guest removal, causing significant Brauner–Emmett–Teller surface area (SA_BET_) reduction from 848 m^2^ g^−1^ to 654 m^2^ g^−1^ [18]. Meanwhile, imine-linked 3D-CageCOF-1, which is constituted by a flexible cage-like triangular prism knot, exhibited a reversible pore structure, switching from large-pore (lp) to small-pore (sp) architectures after the trapped DMF molecules were removed within the pores [19]. Breaking dynamic behavior in 3D COFs is thus pivotal to tackling such structural deformation disadvantages, aiming to achieve high-surface area 3D COFs that are close to their theoretical values, as well as improving the gas storage performance.

The current strategy for breaking dynamic behavior in 3D COFs is by constructing supramolecular forces within the established 3D framework such as steric hindrance and H-bonding to restrict the molecular bond rotation and contraction of the skeleton [20,21]. In this regard, we have reported the design and synthesis of two non-dynamic 3D COFs (JUC-552 and JUC-570) by rationally designing linkers containing alkyls groups as side chains to promote steric hindrance [22,23]. The skeleton of JUC-552 remained less flexible or rigid as the presence of crowded four-symmetric methyl groups restricted the molecular bond rotation of imine linkages, affording an exceptional SA_BET_ (up to 3023 m^2^ g^−1^) compared with that of the non-functionalized 3D COF analogue (JUC-550, SA_BET_ = 846 m^2^ g^−1^) [22]. Likewise, steric hindrance caused by the isopropyl groups in JUC-570 kept the imine linkages less rotatable, thus leading to a nearly fourfold increase in SA_BET_ relative to the unfunctionalized 3D COF analogue (JUC-571) [23]. Meanwhile, the dynamic behavior of dynaCOF-301 was significantly altered as the presence of hydroxyl (–OH) groups near the imine linkages evolved H-bonding, thus enforcing the framework upon removal and inclusion of the guest molecule [24]. Although these strategies obviously restrict or break the dynamic behavior in 3D COFs, the specific design of building units and precise synthetic conditions need to be considered to evolve such supramolecular bonds within the framework. Therefore, the exploration of a new and simpler strategy to tackle the structural dynamic issues in 3D COFs is emergent yet challenging.

Herein, we report a novel yet simple strategy for breaking the dynamic behavior in 3D COFs by the use of a pre-designed conformationally locked linker (Figure 1a). In detail, pre-locked linker-based 3,8-diamino-6-phenylphenanthridine (DPP) was chosen to be condensed with a tetrahedral knot-based tetrakis-(4-formylphenyl)adamantane (TFPA) to afford non-dynamic JUC-595, as the benzylideneamine moiety in DPP locked the linker flexibility and thus restricted the molecular bond rotation of the imine linkages in JUC-595. For a comparison, TFPA was further condensed with a flexible linker-based benzidine (BD) to produce dynamic JUC-594. JUC-594 experienced reversible crystal-phase transformation upon solvent inclusion and release as the impact of framework contraction–expansion, while JUC-595 showed a PXRD profile that was unchanged, confirming the breaking of dynamic behavior. Hence, the activated JUC-595 exhibited a higher SA_BET_ and pore volume than that of JUC-594, manifesting the firmness of the framework upon activation. Accordingly, JUC-595 documented higher CO_2_ and CH_4_ storage compared with JUC-594. Furthermore, JUC-595 recorded a notable normalized H_2_ storage capacity surpassing other reported high-surface area 3D COFs. This work demonstrates facile textural property tuning in 3D COFs and paves the way for the development of functional 3D COFs for gas storage applications.

## 2. Materials and Methods

### 2.1. Materials

Unless specified otherwise, all initial materials and solvents were purchased from J&K Scientific Ltd. (Beijing, China). Reagents and solvents are in high purity and used as received without additional purification.

### 2.2. Synthesis of Dynamic JUC-594

JUC-594 has been prepared elsewhere (termed as DbTd) but with very poor crystallinity [25]; thus, it may be considered as amorphous polymer. JUC-595 in this work, was prepared in a different synthetic condition to obtain highly crystalline 3D COF. TFPA was firstly synthesized according to the reported method: (FT-IR (cm^−1^) *ν*_CH꞊O (aldehyde)_ = 1697, *ν*_CH꞊CH (phenyl)_ = 3029, *ν*_CH2-CH2 (adamantyl)_ = 2928 cm^−1^; ^1^H NMR (400 MHz, CDCl_3_, δ(ppm) = 10.02 (s, 4H), 7.91 (d, 8H), 7.67 (d, 8H), 2.26 (s, 12H) [26]. Specifically, TFPA (0.025 mmol, 13.8 mg) and BD (0.050 mmol, 9.2 mg) were placed into a Pyrex glass tube. Subsequently, 1,4-dioxane (0.75 mL), mesitylene (0.25 mL), and aqueous acetic acid (6 M, 0.1 mL) were added. The tube was flash-frozen in a liquid nitrogen bath and the internal pressure was maintained at 19.0 mbar under vacuum. It was then sealed by flame and reduced in length by ca. 12.0 cm before being placed in an oven at 120 °C for three days to afford a yellow crystalline powder. It was then washed with acetone 5 times and immersed in hexane. Activation was performed by vacuum drying at 26.7 mbar at 65 °C. Elemental analysis, calculated: C: 86.86; H: 5.14; N: 8.00; found: C: 86.56; H: 5.27; N: 8.17.

### 2.3. Synthesis of Non-Dynamic JUC-595

TFPA (0.020 mmol, 11.0 mg) and DPP (0.040 mmol, 11.4 mg) were placed into a Pyrex tube. Subsequently, mesitylene (1 mL) and 6 M aqueous acetic acid (6 M, 0.1 mL) were added into tube. The tube was flash-frozen in a liquid nitrogen bath and the internal pressure was maintained at 19.0 mbar under vacuum. Upon warming to room temperature, the tube was placed in an oven at 120 °C for three days. The resultant precipitate was filtered and washed with acetone 5 times. It was then immersed in hexane. Activation was performed by vacuum drying at 26.7 mbar at 65 °C. Elemental analysis, calculated: C: 88.57; H: 4.76; N: 6.67; found: C: 88.30; H: 4.91; N: 6.79.

### 2.4. Characterization

The crystal model of both COFs was developed using the Materials Studio software (Ver. 8) package [27]. The Fourier transform infrared spectrophotometer (FT-IR) spectra were obtained using a SHIMADZU IRAffinity-1. Thermogravimetric and differential scanning calorimetry analysis (TGA-DSC) were recorded on a SHIMADZU DTG-60 thermal analyzer under N_2_ at 30 °C to 800 °C, with a heating rate of 10 °C min^−1^. The powder X-ray diffraction (PXRD) data were collected on a PANalytical B.V. Empyrean powder diffractometer, using a Cu Kα source (λ = 1.5418 Å) over a range of 2θ = 2.0–40.0° with a step size of 0.02° and 2 s per step. The sorption isotherms for N_2_, CH_4_, CO_2_, and H_2_ were measured using a Quantachrome Autosorb-IQ analyzer with ultra-high-purity gas (99.999% purity). Before gas adsorption measurements, the as-synthesized COFs (~50.0 mg) were immersed in acetone for 12 h (5 × 20.0 mL) and then n-hexane for another 12 h (5 × 10.0 mL). The n-hexane was then extracted under vacuum at 65 °C to afford the samples for sorption analysis. To estimate pore size distributions for both JUC-594 and JUC-595, nonlocal density functional theory (NLDFT) was applied to analyze the N_2_ isotherm on the basis of the model of N_2_@77K on carbon with slit pores. The electron microscopy (SEM) images were acquired using a JEOL JSM-6700 SEM. Meanwhile, the transmission electron microscopy (TEM) images were taken with a JEOL JEM2100F operated at 220 kV.

## 3. Results and Discussion

### 3.1. Design and Synthesis of Dynamic JUC-594 and Non-Dynamic JUC-595

The grand design for breaking dynamic behavior in 3D COFs in this work was to condense a locked biphenyl linker (DPP) to afford rigid and non-flexible 3D COFs (JUC-595, Figure 1). A contrasting phenomenon was expected for a 3D COF with a flexible BD linker (JUC-594), prompting a reversible framework expansion–contraction driven by a rotatable imine bond upon guest removal and inclusion [28]. In practice, both JUC-594 and JUC-595 were synthesized via acid-catalyzed Schiff-based solvothermal condensation among TFPA with either DPP or BD at 120 °C for 3 days. In particular, JUC-594 was re-synthesized with different synthetic conditions in contrast to the previously documented method (DbTd), resulting in a crystalline product with a favorable yield (85%) [25]. JUC-595 was likewise obtained with an 85% yield, reaffirming its high purity.

The FT-IR spectra of both 3D COFs exhibit a stretching band at 1625 and 1626 cm^−1^ for JUC-594 and JUC-595, respectively, indicating the evolution of C=N bonds. Additionally, the absence of an N-H stretching vibration signal from BD (3321 ≈ 3208 cm^−1^) and DPP (3328 ≈ 3212 cm^−1^), as well as the significant diminishment of the C=O vibration signal of TFPA (1695 cm^−1^), confirmed a successful Schiff-based condensation reaction (Appendix A). Furthermore, the solid-state ^13^C cross-polarization magic-angle-spinning nuclear magnetic resonance (ss ^13^C CP/MAS NMR) of the carbon-imine (C=N) peaks identified at 159.2 and 160 ppm for both JUC-594 and JUC-595, respectively, further revealed a successful condensation reaction (Appendix A). Meanwhile, the morphology of both COFs observed under SEM and TEM analysis revealed that both COFs crystallized into homogeneous rod-like crystals (Appendix A).

The crystallinity and unit cell parameters of both COFs were resolved by PXRD analysis in conjunction with structural simulations (Figure 1). The crystal model of both JUC-594 and JUC-595 adopted a two-fold interpenetrated dia net with corresponding cell parameters of a = b = 42.2408 Å, c = 31.5544 Å, and α = β = γ = 90° for JUC-594, and a = b = 44.7519 Å, c = 27.9163 Å, and α = β = γ = 90° for JUC-595 (Figure 1b). The experimental PXRD profile of the fresh JUC-594 exhibited intense peaks at 4.04 and 4.66°, as well as several moderate peaks at 6.52, and 9.40°, corresponding to the (111), (210), (221), and (322) Bragg peaks of the P-4B2 space group, confirming good crystallinity and phase purity (Figure 1). Meanwhile, the reported DbTd showed a broader first peak at 1.26° and other featureless low-intensity peaks, signaling poor crystal quality [25]. Following the same space group, intense PXRD peaks for JUC-595 were observed at 3.91 and 4.38°, with moderate peaks at 6.92, 7.80, and 9.51° belonging to the (200), (210), (112), (321), and (003) Bragg peaks. Full-profile pattern matching (Pawley) refinements were performed based on the experimental and simulated data, revealing good agreement factors (*R_p_* = 3.81%, *R*_wp_ = 5.05% for JUC-594; and *R_p_* = 0.91%, *R*_wp_ = 1.37% for JUC-595). Alternative structures such as non-interpenetrated or multi-fold interpenetrated dia nets were also considered, but the simulated PXRD profiles were inconsistent with the experimental ones (Appendix A). Based on these results, it is proposed that both 3D COFs adopt a two-fold interpenetrated dia net. Furthermore, structural stability is an important parameter for gas storage materials. To access the chemical stability of both JUC-594 and JUC-595, both COFs were immersed into a number of organic solvents, base and acid solutions, as well as in boiling water for 24 h. Notably, both 3D COFs retained their main PXRD peaks even after immersion in various organic solvents (DMF, THF, acetone, EtOH, DCM, and cyclohexane), as well as in boiling water, acid (HCl 0.1 M), and base (NaOH 0.1 M) for 24 h, signaling good chemical stability (Appendix A). Similarly, there was no noticeable alteration in their FT-IR spectra following immersion in these solvents, indicating that there were no changes in functional or chemical composition as a result of this treatment (Appendix A). Furthermore, TGA analysis confirmed that both COFs were thermally stable up to 450 °C under N_2_ atmosphere (Appendix A). Additionally, their DSC features clearly revealed the distinct thermal characteristic of both COFs, in which an obvious earlier endothermic point at 187 °C was observed for JUC-594, which was associated with the possible initial destruction of the framework and the release of the trapped solvent molecules, while JUC-595 did not show a similar feature. This result may be associated with the solvent-triggered structural deformation in JUC-594.

### 3.2. Structural Dynamic Behavior of JUC-594 and JUC-595

It has been observed that the dynamic behavior of porous materials could induce reversible crystal transition due to the framework contraction–expansion caused by a guest molecule, pressure, or temperature [18,29,30,31]. In this work, we propose a pre-locked linker strategy for breaking dynamic behavior in 3D COFs. Firstly, the PXRD profiles of both COFs under solvated (hexane) and activated conditions were recorded (Appendix A). Obviously, JUC-594 shows PXRD profile discrepancy between the solvated and activated samples, meaning that both samples have a different crystal phase (Appendix A). Meanwhile, the PXRD profiles of JUC-595 depict a similarity for both solvated and activated conditions, confirming that the inclusion and exclusion of solvent does not affect the framework (Appendix A). To obtain insights into this guest molecule-triggered crystal transition, PXRD profiles of both COFs upon evaporation of the trapped hexane at different times were further recorded to observe their structural dynamic responsivity (Figure 2). Apparently, the peaks at 2θ = 4.66 and 4.04° of solvated JUC-594 gradually merged into an intense peak at 4.07° upon gradually evaporating hexane in the pores within 1–20 min in room temperature (Figure 2a). This quick PXRD peak change identifies the structural dynamic behavior responsive in JUC-594, leading to framework unit cell expansion along the *a*- and *b*-axes upon guest molecule release. Interestingly, the former two peaks re-split upon being resoaked in hexane, confirming the framework contracting upon solvation. Hence, JUC-594 show reversible structural dynamic behavior. Unlike the framework contraction upon activation found in dynamic FCOF-5 and LZU-301 with a non-bulk tetraphenylmethane knot [18,30], we assumed that the presence of a hydrophobic hexane molecule in JUC-549 may attract pore surface-procuring framework contraction. More importantly, the bulkier TFPM knot may cause a less dynamic character in JUC-594 compared with FCOF-5 and LZU-301.

In contrast, PXRD profiles of JUC-595 remained unchanged upon hexane inclusion and release from the pores (Figure 2b). This observation indicates that JUC-595 does not undergo crystal transformation caused by hexane molecules, as the framework remains rigid and flexible. As we proposed, the intramolecular locking by benzylideneamine moieties in DPP could cause molecular bond rotation restriction on imine linkages, breaking the dynamic behavior in JUC-595. The strategy to restrict the molecular bond rotation of the 3D COFs in this present work is thus novel compared with the previous work, with steric hindrance and H-bonding strategies [20,21].

The structural dynamic in 3D COFs greatly affects pore character, thus determining surface areas and pore volume. In this regard, N_2_ adsorption–desorption analyses were conducted at 77 K to evaluate the permanent porosity of both 3D COFs (Figure 3 and Appendix A). Both COFs exhibited type-I isotherms, with a sharp uptake observed below P/P_0_ = 0.05, characteristics of microporous materials (Figure 3a). Comparatively, based on these isotherms, the activated JUC-595 recorded a higher Brunauer–Emmett–Teller specific surface area (BET SSA) (754 m^2^ g^−1^) in comparison with the activated JUC-594 (548 m^2^ g^−1^) (Appendix A). The lower SSA of JUC-594 may be attributed to the framework contraction upon activation due to its structural dynamic behavior. A similar surface area reduction as a result of this structural dynamic was also reported in flexible JUC-550 (846 m^2^ g^−1^) relative to its structural analog (JUC-552, BET SSA = 3023 m^2^ g^−1^), as well as in flexible JUC-571 relative to JUC-571 [22,23]. Additionally, the N_2_ adsorption isotherm feature of JUC-594 displays an obvious hysteresis loop, indicating a structural dynamic response upon N_2_ molecules’ release from the pore (Appendix A) [18]. Furthermore, both COFs showed distinct pore size distributions calculated by a nonlocal density functional theory (NLDFT) method (Figure 3b,c). JUC-594 afforded a dominant pore size of about 1.09 nm, with a pore volume of 0.247 cm^3^ g^−1^, narrower than that of JUC-595 (dominant pore size = 1.46 nm, pore volume = 0.553 cm^3^ g^−1^). Meanwhile, the predicted pore size of simulated non-contracted JUC-594 is wider (2.2 nm) than that obtained from the experimental result, revealing the N_2_ molecule-triggered pore change (Appendix A). In contrast, JUC-595 confirmed a comparable pore size between the experimental and simulated non-contracted framework (1.6 nm), strengthening the non-dynamic framework upon guest molecule inclusion–exclusion (Appendix A).

### 3.3. Gas Storage Performance of JUC-594 and JUC-595

Having observed that tuning the structural dynamic affects the porosity of both 3D COFs, we then sought to study their CO_2_, CH_4_, and H_2_ storage performances. In pursuit of this, the CO_2_ and CH_4_ gas storage of JUC-594 and JUC-595 at 273 and 298K were firstly compiled, as these gases possess a comparable molecular size (Figure 4). At a lower temperature (273 K), JUC-594 recorded a CO_2_ adsorption capacity as high as 34 cm^3^ g^−1^ and experienced storage decline down to 22 cm^3^ g^−1^ at 298 K (Figure 4a,b). Comparatively, JUC-595 exhibited higher adsorption capacities in similar conditions (44 and 30 cm^3^ g^−1^ at 273 and 298 K, respectively). Similar trends were observed for CH_4_ as a gas probe, in which JUC-595 stored as high as 11 cm^3^ g^−1^ of the gas at 273 K and 7 cm^3^ g^−1^ at 298 K, slightly surpassing the performance of JUC-594 (9 and 5 cm^3^ g^−1^ at 273 and 298 K, respectively) (Figure 4c,d). We assumed that the increased adsorption capacities of JUC-595 compared with JUC-594 for both gas probes correlates to their distinct porosity characters as the impacts of their dissimilar structural dynamic behavior.

Considering the highly valuable and renewable H_2_ as a cost-effective fuel alternative to fossil fuel in the future [32], the potential H_2_ storage of JUC-594 and JUC-595 was further investigated. In particular, JUC-594 exhibited an H_2_ adsorption capacity as high as 55 cm^3^ g^−1^ at 77 K and 38 cm^3^ g^−1^ at 87 K (Figure 5a). Notably, JUC-595 reached an H_2_ adsorption capacity of 96 cm^3^/g^−1^ at 77 K and 65 cm^3^ g^−1^ at 87 K, which are nearly double that of JUC-594 under identical test conditions (Figure 5b). To gain a better understanding of the improved H_2_ storage in JUC-595, the H_2_ isosteric heat adsorption (*Q_st_*) was calculated based on H_2_ adsorption tests at different temperatures. Apparently, the calculated *Q_st_* for JUC-595 was about 6.57 kJ mol^−1^, higher than that of JUC-594 (5.36 kJ mol^−1^) (Figure 5c,d). These findings suggest that the interaction of H_2_ with JUC-595 is stronger compared with JUC-594. The enhanced sorbate–sorbent interaction in JUC-595 is likely attributed to the presence of more adsorption-active sites, as well as the non-dynamic frameworks maintaining the surface area and pore volume. More importantly, the surface area-normalized H_2_ sorption capacity of JUC-595 at 77 K (0.13 cm^3^ m^−2^) is far superior in comparison with other reported 3D COFs, including COF-103 (0.04 cm^3^ m^−2^) [33], JUC-596 (0.12 cm^3^ m^−2^) [34], and JUC-597 (0.08 cm^3^ m^−2^) [34], and is comparable with that of the highly connected 3D JUC-568 (0.19 cm^3^ m^−2^) [35] under similar conditions (Table 1).

## 4. Conclusions

In summary, we used a pre-locked linker as a novel strategy for breaking dynamic behavior in 3D COFs. Compared with flexible BD, the benzylideneamine moiety in DPP conformationally locks the linker and greatly restrict the molecular bond rotation of imine linkage in JUC-595. This strategy afforded a rigid 3D framework even under solvent inclusion and release, while JUC-594 exhibited reversible transformation driven by its flexible imine bonds. Beneficially, an improved surface area was seen in JUC-595 compared with JUC-549. Accordingly, JUC-595 recorded enhanced CO_2_, CH_4_, and H_2_ storage relative to JUC-594. This study serves as an important insight into breaking the dynamic behavior in 3D COFs affecting gas storage performance.

## Data Availability

Data are contained within the article and Appendix A.

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
