# Peer review of "Breaking Dynamic Behavior in 3D Covalent Organic Framework with Pre-Locked Linker Strategy"

_nanomaterials, 2024, doi:10.3390/nano14040329_

Round 1

Reviewer 1 Report

Comments and Suggestions for Authors

The manuscript by Chen et. al. reported fabrication of dynamic JUC-594 and non-dynamic JUC-595 via dynamic imine condensation of TFPA with BD and rigid DPP respectively. Furthermore, the frameworks phase purity, porosity, thermal stability, dynamic functionalities, gas adsorption were analyzed by PXRD, surface area, TGA, FT-IR respectively. This work is of interest to readers and could be suitable for Nanomaterials, MDPI after consideration of the following points. 

1. In addition to FT-IR, the authors should include solid state 13C NMR to confirm the presence of imine functionalities in JUC-594 and JUC-595. 

2. The authors should investigate the stability of JUC-594 and JUC-595 in protic solvents, aprotic solvents, and aqueous solution via PXRD and FT-IR analysis. 

3. The authors should cite the following articles in the introduction section; Membranes 2023, 13, 696;  Commun Chem. 2018, 1, 98.  

Comments on the Quality of English Language

Minor editing of English language required. 

Reviewer 2 Report

Comments and Suggestions for Authors

The paper "Breaking Dynamic Behavior in 3D Covalent Organic Framework with Pre-locked Linker Strategy" reports on the preparation, structural characterization and evaluation of gas storage properties of two 3D COFs constructed via azomethine linkage starting from two diamines as building blocks with structural flexibility/ rigidity.

The manuscript is of interest for the field of COFs. The experimental data are briefly presented with more emphasis on the structural dynamic behavior.

The Introduction section is too general, with some previously reported results, but it is not clear enough whether the dynamic behavior is considered an advantage or a disadvantage for COFs. So, I consider that some improvements on the introduction is necessary.

Section 2. The Materials used in the preparation of COFs must be presented.

Sections 2.2. and 2.3 - the main FT-IR bands must be specified. Also, the results of NMR analysis, at least 1H NMR.

Section 3.1. It is not clear if these structures are new or the syntheses are reproduced from literature. The novelty must be highlighted as: the purity of the compounds, the yield, the crystallinity degree, etc.

FT-IR analysis is too briefly presented as it is the main structural analysis used. DSC analysis must be also used for thermal characterization of the compounds.

PXRD data must be compared with the literature reported structures.

Section 3.2. Some literature comparisons must be added and discussed.

Based on these observations I recommend Major revision of this paper before acceptance.

Reviewer 3 Report

Comments and Suggestions for Authors

The authors have described the synthesis of imine-linked 3D COFSs (JUC-552 and JUC-570) via breaking dynamic behavior. It is probably interesting work but the novelty is not clear enough as the authors have synthesized crystalline JUC materials compared to the previous research in literature. Therefore, the authors are suggested to improve the manuscript based on the comments below:

 The obtained results should be discussed and compared with literature more in details.

Please explain this term: “Anal. Cald” and “Found” in section 2.2 and 2.3.

Please use one unit for gas pressure: mTorr or mbar.

What was the working voltage of TEM in this work?

Some texts in result and discussion belongs to material and methods section and should be moved. For example, “using Materials Studio software package” is for the experimental section. Please revise your manuscript based on this suggestion.

Figure 1: What are the peaks between 10 and 25° ?

Figure 2: please indicate every peaks in these graphs.

Too much graphs are moved to supporting information which is not attractive enough as in supporting information, the images are not clean and clear enough. For example, the scale bars of both SEM/TEM images are not readable.

Comments on the Quality of English Language

English is still OK and readable. 

Round 2

Reviewer 2 Report

Comments and Suggestions for Authors

The authors have responded to all the suggestions and the paper can be accepted in the revised form.

Reviewer 3 Report

Comments and Suggestions for Authors

The authors have improved the manuscript based on reviewers' comments. It should be suitable for the publication.

Comments on the Quality of English Language

OK